# Soil Zinc Is Associated with Serum Zinc But Not with Linear Growth of Children in Ethiopia

**DOI:** 10.3390/nu11020221

**Published:** 2019-01-22

**Authors:** Masresha Tessema, Hugo De Groote, Inge D. Brouwer, Edith J.M. Feskens, Tefera Belachew, Dilnesaw Zerfu, Adamu Belay, Yoseph Demelash, Nilupa S. Gunaratna

**Affiliations:** 1Division of Human Nutrition and Health, Wageningen University, 6700 AA Wageningen, The Netherlands; inge.brouwer@wur.nl (I.D.B.); edith.feskens@wur.nl (E.J.M.F.); 2Food Science and Nutrition Research Directorate, Ethiopian Public Health Institute, Gulele Sub City, 1242 Addis Ababa, Ethiopia; dilnesaw2012@gmail.com (D.Z.); adamu_bel2000@yahoo.com (A.B.); yose.deml@gmail.com (Y.D.); 3Human Nutrition Unit, Jimma University, P.O. Box 378, Jimma, Ethiopia; teferabelachew@gmail.com; 4International Maize and Wheat Improvement Centre (CIMMYT), 1041-00621 Nairobi, Kenya; h.degroote@cgiar.org; 5Department of Nutrition Science and Public Health Graduate Program, Purdue University, West Lafayette, IN 47907, USA; gunaratna@purdue.edu

**Keywords:** soil zinc, serum zinc, linear growth, soil fertility, preschool children, Ethiopia

## Abstract

To our knowledge, the relationships among soil zinc, serum zinc and children’s linear growth have not been studied geographically or at a national level in any country. We use data from the cross-sectional, nationally representative Ethiopian National Micronutrient Survey (ENMS) (*n* = 1776), which provided anthropometric and serum zinc (*n* = 1171) data on children aged 6–59 months. Soil zinc levels were extracted for each child from the digital soil map of Ethiopia, developed by the Africa Soil Information Service. Children’s linear growth was computed using length/height and age converted into Z-scores for height-for-age. Multi-level mixed linear regression models were used for the analysis. Nationally, 28% of children aged 6–59 months were zinc deficient (24% when adjusted for inflammation) and 38% were stunted. Twenty percent of households in the ENMS were located on zinc-deficient soils. Soil zinc (in mg/kg) was positively associated with serum zinc (in µg/dL) (b = 0.9, *p* = 0.020) and weight-for-height-Z-score (b = 0.05, *p* = 0.045) but linear growth was not associated with soil zinc (*p* = 0.604) or serum zinc (*p* = 0.506) among Ethiopian preschool children. Intervention studies are needed to determine whether there are causal links between soil and human zinc status.

## 1. Introduction

Zinc is an essential micronutrient for both plants and animals, including humans. It is estimated that about 17% of the global population has inadequate zinc intake [1] and zinc deficiency is widespread in developing countries [2]. Zinc supports normal growth and development during pregnancy and childhood and it is required for the catalytic activity of approximately 100 enzymes; it plays a role in immune function, protein synthesis, wound healing, DNA synthesis and cell division [3,4,5]. Zinc deficiency is caused mainly by insufficient intake or inadequate absorption of zinc in the body [6]. Human zinc deficiency is highly prevalent in sub-Saharan Africa, where diets are typically high in cereals and low in animal source products and contain low levels of bioavailable zinc [7,8]. Studies on the effect of zinc supplementation on linear growth of children showed conflicting results [9,10,11]. For instance, a review in developing countries found that zinc supplementation has a significant effect on linear growth of children [9]. However, another review found that zinc supplementation did not have a significant effect [11]. A recent systematic review published in Cochrane suggested that zinc supplementation resulted only in a marginal improvement in linear growth of children [10]. An earlier study conducted with Ethiopian preschool children found that zinc supplementation significantly improved linear growth of stunted children [12]. The recent Ethiopian National Food Consumption Survey demonstrated that over half of preschool children in Ethiopia are estimated to have low dietary zinc intake [13].

Zinc is also important for plants, including food crops. Soils with insufficient zinc for optimal crop growth are classified as zinc deficient. Zinc deficiency in agricultural soils is a global problem reported in many countries [14,15]. Most soils in sub-Saharan Africa are affected by zinc deficiency [16,17]. Soil zinc deficiency has a major effect on food security and human health by limiting the yields and the grain zinc concentrations of staple crops grown on zinc-deficient soils [10,15], especially in Africa [16,17,18,19,20]. Several studies have shown that zinc fertilizers, applied to the soil or through the foliar application, improved both the zinc content and yield of grains [16,21,22,23,24]. The relationships among soil zinc, serum zinc and linear growth of children are poorly understood [25].

To our knowledge, to date, no study has established a quantitative relationship among soil zinc, human serum zinc status and linear growth of children from nationally representative data. We hypothesize that a lower soil zinc level is associated with lower grain zinc levels and lower zinc levels in the diet, resulting in higher prevalence of linear growth failure among preschool children, mediated by lower serum zinc status (Figure 1). This would support emerging efforts to improve human zinc status by improving soil zinc status [16]. Improved soil zinc could also increase crop productivity and production, which in turn could improve children’s growth through higher incomes, lower food insecurity, reduced inflammation through improvements in the health environment and increased resources for child feeding and caregiving (Figure 1). Using data from two nationally representative cross-sectional surveys on soils and children’s nutritional status, this is the first study that assesses the geographical distribution of poor zinc soils, poor serum zinc status and poor linear growth and their relationship among Ethiopian preschool children.

## 2. Materials and Methods

### 2.1. Study Design and Study Population

We merged two datasets: the Ethiopian National Micronutrient Survey (ENMS), which provided serum zinc and anthropometric data from children in georeferenced households and the Africa Soil Information Service (AfSIS) soil map, which provided soil zinc levels as raster data [27]. The ENMS was designed as a regionally- and nationally-representative cross-sectional survey of children (6–59 months) and was conducted between March and July 2015. Ethiopia is administratively sub-divided into nine regional states and two city administrations (Addis Ababa and Dire Dawa) [28]. The ENMS enumeration areas (EAs) or clusters are geographic areas defined by the Central Statistics Agency (CSA) for the Ethiopia Population and Housing Census [28]. EAs contain on average 181 households (150 to 200) and are subsets of the regions [28].

In the ENMS, 366 clusters were first randomly selected from all region or city administrations with probability proportional to size. Prior to the actual survey, all households within the boundary of each selected cluster were listed and a census was conducted of the people living in the households. In the next stage, within each selected cluster (or segment of clusters), 11 households were randomly selected. In the final stage, all preschool children aged 6 to 59 months in the 11 households were selected for the actual survey. If eligible occupants of a house were not present, two return visits with a written appointment were made. If no eligible respondents were available during the two visits, the household was recorded as refusing to participate and was not replaced. The children’s mother or caretaker responded to the questionnaires on behalf of the children. In addition to blood samples, anthropometric measurements of the children were collected. Further, information relating to the household’s demographic and socioeconomic characteristics and their geographic coordinates were obtained. A total of 4026 households in 366 clusters across the nine regions and the two administrative cites were selected; about 92% (*n* = 3700) of households in 353 clusters gave their consent and were included in the study. In the consenting households, 1776 preschool children (on average about 5–6 children per EA) were eligible for blood collection and adequate blood samples for serum zinc were collected from 1171 children in 316 clusters. Anthropometric data were collected from 1673 children. To account for the multistage sampling employed in the ENMS, sample weights were calculated from the 2007 census [28] and used to estimate regional and national prevalence of households on zinc-deficient soils and of serum zinc deficiency and stunting among children under five years of age [29].

Ethical clearance was obtained from the National Research Ethical Review Committee of the Ethiopian Science and Technology Ministry (number 3.10/433/06). Informed consent was obtained from all adults who were interviewed, specifically the household head and caregiver.

### 2.2. Data Collection and Analysis

#### 2.2.1. Collection, Processing and Analysis of Biochemical Samples

In the ENMS, non-fasting venous blood was collected aseptically in the morning by experienced phlebotomists from the left arm by venepuncture using vacutainer trace element-free tubes (Royal Blue top tube, 6.0 mL). The samples were collected at the household, placed in cold boxes containing frozen gel packs (<8 °C) and transported as soon as possible after collection to the centralized temporary field laboratory sites. Within ~1 h of collection, the blood was allowed to clot for 30 min and centrifuged at 3000 rotations per minute (rpm) for 10 min. An aliquot was separated based on the recommended procedures of the International Zinc Nutrition Consultative Group [30].

The zinc status of under-five children was assessed by serum zinc, which is the recommended biomarker to estimate zinc status [31]. Serum zinc concentration was measured using Shimadzu Flame Atomic Absorption Spectroscopy (AA 6800 Japan model) with an air-acetylene flame at a wavelength of 213.9 nm and a slit width of 0.7 nm. Serum zinc deficiency was defined as concentration < 65 µg/dL for children [30]. Staff serum samples were used as a control during analysis of every 30 samples and intra-assay CV was 4.3%. Inflammation was measured using C-reactive protein (CRP) and α-1-glycoprotein protein concentration (AGP). CRP and AGP concentrations were determined using the fully automated Cobas 6000 immune-turbidimetry method using Roche kits (Roche Diagnostics, GmbH, Mannheim, Germany) [32]. Further, diarrhoea was defined as three watery or loose stools in any 24-h period during those two weeks and measured by asking the caregiver to recall diarrhoea incidence in the two weeks prior to the survey.

Serum zinc concentration was adjusted for inflammation using the biomarkers CRP and AGP, according to the regression correction method proposed by the BRINDA Working Group [33]. Reference concentrations (maximum of lowest decile) for serum CRP and AGP were used to avoid over-adjusting serum zinc among preschool children with low levels of inflammation [32,34]. Acute or chronic inflammation were defined by serum CRP > 5 mg/L or AGP > 1 g/L, respectively [35]. Stages of inflammation were categorized as no inflammation (CRP ≤ 5 mg/L and AGP ≤ 1 g/L); incubation (CRP > 5 mg/L and AGP < 1 g/L); early convalescence (CRP > 5 mg/L and AGP > 1 g/L); late convalescence (CRP ≤ 5 mg/L and AGP > 1 g/L) [35]. The analyses of serum zinc and inflammation biomarkers were conducted at the Ethiopian Public Health Institute laboratory, certified by the Ethiopian National Accreditation Office in accordance with the requirements of ISO 17025:2005 and ISO 15189:2012.

#### 2.2.2. Demographic and Socioeconomic Characteristics

A three-week training course was provided for the ENMS data collectors and supervisors on data collection and overall quality control, followed by one week of field pilot testing in a cluster not selected for the survey. After the pilot testing, the questionnaires were revised before the actual survey. Demographic characteristics collected included family size and sex and age of children; socioeconomic characteristics included household assets, household food insecurity status and education level of the children’s caretaker. In addition, data were collected on the foods (animal and plant-based) consumed by children in the last 24 h [36]. Household food insecurity status was measured using the validated three-month food insecurity experience scale (FIES) [37].

#### 2.2.3. Anthropometric Data

Anthropometrics (i.e., weight and height or recumbent length) were collected on the selected children. Their weight was measured with light clothing and without shoes to the nearest 100g using a standard UNICEF SECA 874 U digital scale (UNICEF Supply Division, Copenhagen, Denmark). The scale was calibrated using a standard weight after moving from one household to the next. The length of younger children (6–23 months) was measured in a recumbent position to the nearest 0.1cm using a UNICEF measuring board (UNICEF Supply Division, Copenhagen, Denmark) with an upright wooden base and a movable headpiece. The height of children older than 23 months of age was measured in a standing position to the nearest 0.1 cm. All anthropometric measurements were taken twice (or three times if the measurements differed between the first and the second reading) and the average values were taken. The age of the children was calculated based on the date of birth and the date of the interview. The weight and height/length of the children were converted into Z-scores for height-for-age (HAZ) and weight-for-height (WHZ) according to 2006 WHO child growth standards, using WHO Anthro software [38]. Linear growth failure was computed using HAZ; HAZ scores less than two standard deviations below median values were considered indicative of stunting.

#### 2.2.4. Collection and Analysis of Soil Zinc

The soil zinc map of Ethiopia was obtained from AfSIS. The map provides a grid of 1 km^2^ and for each grid cell a soil zinc level in mg/kg. The methodology used to obtain soil zinc data and derive the map have been described elsewhere [27,39]. Soil zinc levels were extracted for all households based on their geographic coordinates and merged with the ENMS data. Zinc-deficient soils were defined as having zinc levels lower than 1.5 mg/kg [40].

### 2.3. Statistical Analysis

Statistical analyses were conducted with SAS version 9.3 (SAS Institute, Cary, North Carolina, USA). A wealth index was created using the first principal component [41] constructed with the following household assets (binary variables): grid electricity, watch, radio, television, mobile telephone, landline telephone, refrigerator, solar panel, bicycle, motorcycle, animal-drawn cart, car and motorboat. Using this index, households were assigned to wealth tertiles.

Spearman’s rank correlation was used to investigate correlations among soil zinc, child’s serum zinc, child’s growth (HAZ and WHZ scores), child’s inflammation markers (CRP and AGP) and household food insecurity. To examine the associations among soil zinc, serum zinc and linear growth of children, a multi-level mixed linear model with a random intercept was fitted with restricted maximum likelihood estimation using the SAS procedure “proc mixed.” Clusters were used as a random intercept. Independent variables in the model were those known or suspected to be biologically important predictors of child growth or serum zinc (Figure 1). Model diagnostics were checked to ensure assumptions of normality of error terms and homogeneity of error variance were met.

## 3. Results

### 3.1. Population Characteristics

The median (25th, 75th percentile) age of the 1776 children in the ENMS was 36 (24, 48) months and 48% were female (Table 1). One in seven children had diarrhoea in the two weeks prior to the study. About one-third of the children with diarrhoea received medication during diarrheal illness. Consumption of meat and meat products was minimal (11%) in the 24 h preceding the survey. The median (25th, 75th percentile) serum zinc concentration was 74 μg/dL (63.4, 87.4) indicating relatively low serum zinc values in this population.

A substantial proportion of households living in the ENMS were found to be located on zinc-deficient soils (20%). The prevalence of households living on zinc-deficient soils varied between the administrative regions and in general was higher in the lowlands of Ethiopia and in sparsely populated regions. Among the populous regions in the highlands of Ethiopia, more households were located on zinc deficient soils in Tigray (50%) and Amhara (25%). However, the prevalence of households on zinc deficient soils was lower in Southern Nations, Nationalities and Peoples’ Region (SNNPR) (2%) and Oromia (17%).

The prevalence of serum zinc deficiency was comparable across age and sex groups (20–25%) (Table 2). The highest adjusted prevalence of serum zinc deficiency was found in Afar (34%), Tigray (29%), Amhara and Harari (28%); and the lowest in Gambella (11%) and Benishangul (16%). Adjustment for inflammation decreased the overall prevalence of zinc deficiency from 28% to 24% (Table 2).

The national prevalence of growth failure or stunting was 38%, being higher in rural areas (39%), in boys (41%), in Tigray (44%) and in Amhara (42%) and lower in Addis Ababa (16%) and Gambella (21%). Stunting prevalence was higher in older age groups (Table 2). The geographic distribution of stunting, poor zinc soil and poor serum zinc status is indicated in Figure 2. Linear growth failure and zinc deficiency in preschool children were prevalent in all regions. Further, most lowland regions of Ethiopia were affected by low soil zinc status (Figure 2).

### 3.2. Correlations among Soil Zinc, Serum Zinc, Inflammation and Children’s Growth

Soil zinc level was significantly correlated with serum zinc level (*r* = 0.09, *p* = 0.006), WHZ (*r* = 0.08, *p* = 0.001), inflammation biomarkers: CRP (*r* = 0.07, *p* = 0.017) and AGP (*r* = 0.12, *p* < 0.001) and diarrhoea (*r* = 0.06, *p* = 0.027). Serum zinc concentration was significantly negatively correlated with inflammation (AGP) (*r* = −0.08, *p* = 0.010). WHZ was significantly negatively correlated with household food insecurity (*r* = −0.07, *p* = 0.009) and diarrheal illness (*r* = −0.07, *p* = 0.005). HAZ was not correlated with soil zinc level or serum zinc status (Table 3).

### 3.3. Associations among Soil Zinc, Serum Zinc and Child Growth

#### 3.3.1. The Association between Soil Zinc and Serum Zinc

Using a multi-level mixed linear regression model to predict serum zinc (*n* = 1171), we found that soil zinc level was positively associated with child’s serum zinc concentration (b = 0.9 μg/dL, *p* = 0.019), indicating that a 1 mg/kg increase in soil zinc content was associated with an increase in child serum zinc level of 0.9 μg/dL (Table 4). A quadratic effect of soil zinc on serum zinc was not significant (result not shown). Other factors that significantly negatively affected serum zinc status were inflammation level (CRP and AGP) (b = −6.8, *p* = 0.0008) and time since most recent meal (b = −6.3, *p* = 0.057) (Table 4). We have conducted subgroup analyses by age of children (≥24 months and ≤23 months) and residence (rural and urban). Our subgroup analyses by age group found that soil zinc was significantly and positively associated with serum zinc in children ≥24 months (b = 0.8 μg/dL, *p* = 0.043) (Appendix A), who consume more foods and are less dependent on breastmilk. However, soil zinc was not significantly associated with serum zinc in children ≤23 months (b = 1.8 μg/dL, *p* = 0.099) (Appendix A). Soil zinc was not significantly associated with height-for-age in either age group (Appendix A). We also found that soil zinc was significantly and positively associated with serum zinc from children in both urban and rural areas (Appendix A).

#### 3.3.2. The Association between Soil zinc and Child Growth

The results of the multi-level mixed linear regression models to predict linear growth of children (HAZ) show that soil zinc level was not associated with HAZ (*p* > 0.05) (Table 5, model 1). Other factors such as inflammation level, wealth status and food security status were also not associated with HAZ (*p* > 0.05). Further, adjustment for serum zinc had no significant effect on the relationship between soil zinc and child growth (Table 5, model 2).

A similar model, now with WHZ as the dependent variable, found WHZ to be positively associated with soil zinc (b = 0.05, *p* = 0.026); the association remained when serum zinc was included in the model (Table 6, models 1 and 2). The children with lower wealth status had lower WHZ than children from relatively wealthier households (b=−0.40, *p* = 0.001) and this association remained after adjusting for serum zinc status (b = −0.43, *p* = 0.0002).

## 4. Discussion

We found that the risk of zinc deficiency as measured by low serum zinc concentrations was high among Ethiopian preschool children. This underlines the fact that zinc deficiency in Ethiopia is a public health problem. Soil zinc was positively associated with children’s serum zinc. The associations persist even when controlling for other factors. To the best of our knowledge, this is the first study to investigate quantitatively the association between soil zinc level, serum zinc status and linear growth in Ethiopian children using data from nationally representative studies.

A high proportion of ENMS households were located on zinc-deficient soils, although this varied among regions. We found that in sparsely populated regions such as Gambela, many households lived on soils deficient in zinc. Those low altitude areas are mostly pastoral or semi-agrarian. Furthermore, both soil and serum zinc deficiencies were high in populous regions such as Tigray and Amhara but other populous regions such as Oromia and SNNPR had low levels of soil zinc deficiency. This may suggest that soil in agrarian areas in Ethiopia is more fertile or nutrient-rich than pastoral or non-agrarian regions. Furthermore, these findings suggest that high prevalence of soil zinc deficiency in some regions such as Tigray and Amhara may contribute to lower agricultural productivity and food insecurity, which can result in high zinc deficiency and poor child growth.

The results support our hypothesis of a dietary mechanism in which people grow crops on soils deficient in zinc or with sufficient zinc and consume these crops, affecting serum zinc either negatively or positively (Figure 1). Specifically, the present study may suggest that low soil zinc lowers production or crop yield and the zinc content of grain, which in turn lowers zinc intake and serum zinc levels. Low soil zinc levels have been shown to decrease yields of major crops and therefore reduce agricultural production [24]. An earlier review covering 10 African countries demonstrated that the application of zinc in soil or by foliar fertilization increased the median Zn concentration of maize, rice and wheat grain [16]. Other developing countries in Africa are experiencing lower food production per capita as a result of unhealthy soils and a loss of soil nutrients [17,18]. A recent review showed that zinc deficiency in agricultural soils limit crop production, with yield losses up to 40% [40]. Further, a recent study from China showed that agronomic zinc biofortification (zinc fertilizer on the soil) of wheat greatly increased grain zinc content and improved the zinc bioavailability in grain and flour [19]. The current plan and initiative by the Government of Ethiopia to address soil nutrient deficiency with blended fertilizers should be implemented as soon as possible [42]. This is likely to increase crop yields as well as the micronutrient content of crops grown on zinc-deficient soils, with the potential of reducing human zinc deficiency among the people living in those areas.

The linear growth of children in our study was not associated with soil zinc level and serum zinc status. Therefore, we were unable to detect a mediation effect of serum zinc on the relation between soil zinc and children’s growth. In contrast, an intervention study in Ethiopian preschool children demonstrated that zinc supplementation significantly improved linear growth of stunted children [12]. Several national surveys in other countries showing that serum zinc was not associated with linear growth in children [43,44,45]. The causative mechanisms for childhood linear growth failure are still poorly understood [46] but existing evidence shows that the cause of linear growth failure is multifactorial [46]. Our findings suggest the need for a longitudinal and interventional study to understand causal linkages between soil zinc, serum zinc and linear growth in children.

We found that the risk of zinc deficiency based on low serum zinc concentrations was high, with high variability among regions. Nationally, 28% of children were deficient in zinc, reducing to 24% when adjusted for inflammation, confirming that inflammation causes an overestimation of the prevalence of zinc deficiency [32]. While the correlation between serum zinc and soil zinc was significant, it was relatively low (*r* = 0.09), indicating that other determinants affect serum zinc, including inflammation and phytate content of complementary foods. The relationship between serum zinc and inflammation has not been widely studied [31]. In this study, the correlation between inflammation and serum zinc was also relatively low (*r* = −0.08 with AGP). However, our adjusted regression analysis showed that the effect of inflammation on serum zinc was high (Table 4). We included other factors in these models, such as meat consumption and time since the last meal, along with child and household characteristics but these factors did not have significant effects. Earlier studies, both experimental studies with animal models and human studies of infected and non-infected adults, indicate that systemic infections producing an acute phase response cause the plasma zinc concentration to fall [47]; this is in line with the findings in our current study. Therefore, to determine zinc status in populations, inflammation should be taken into account, which will likely lead to a substantial decrease in the estimates of zinc deficiency prevalence. Health promotion and disease prevention programs may be considered as complementary strategies to reduce zinc deficiency in Ethiopia. Another factor is the effect of phytate on zinc bioavailability, which has not been addressed in our study. However, the existing evidence in Ethiopia and other developing countries show that phytate concentrations are high in cereals-based complementary foods and may inhibit zinc absorption [48]. The Ethiopian National Food Consumption Survey indicated that phytate intake from children’s complementary foods was high, with low variability at the sub-national level [13]. Strategies to reduce phytate in children’s complementary foods may therefore also be considered as a strategy to reduce zinc deficiency in Ethiopia.

In the absence of a gold standard biomarker for zinc status, plasma or serum zinc is endorsed to be the best available biomarker of zinc status [49] for both zinc exposure and the risk of clinical deficiency [50]. Serum zinc is associated with dietary zinc intake, responds consistently to zinc supplementation and decreases with very low zinc intakes [50]. It is suggested that in nutritionally deficient children like in our study population, a higher sequestration rate of zinc by tissues in need of zinc may lead to a higher functional response [51]. However, there are limitations in using serum zinc to assess zinc status: serum zinc responds less to additional zinc provided in food than to a supplement administered between meals, serum zinc seems to predict functional responses to supplementation only when the initial serum zinc concentration is very low, there is large interindividual variability in serum zinc with changes in dietary zinc, and serum zinc is influenced by recent meal consumption, the time of the day, inflammation and certain drugs and hormones [50]. Further research is needed to evaluate potentially useful biomarkers such as hair, nail, or urinary zinc.

Strengths of our study include large population coverage based on nationally representative samples. We used advanced analysis methods to elucidate the association among soil zinc level, serum zinc status and linear growth of children in Ethiopia. Exposure (to poor soil) and outcomes (serum zinc and child growth) were measured on the same individuals or intrapolated from a grid based on their georeference and used to analyse relationships among these variables while adjusting for potential confounders. Despite the strengths of the current study, it also has limitations; in particular, it did not measure all factors that may be important for children’s linear growth. Further, because of the cross-sectional design of our study, we were unable to draw conclusions about the causal effect that low soil zinc might have on lower serum zinc and poor linear growth in children.

## 5. Conclusions

In conclusion, low agricultural soil zinc was found to be a predictor of lower serum zinc and lower weight-for-height among Ethiopian preschool children but neither soil zinc nor serum zinc were associated with linear growth of preschool children in Ethiopia. The relationship between soil and serum suggests that interventions to improve soil zinc fertility could benefit children with zinc deficiency in rural areas, especially those reliant on subsistence agriculture. This could be an alternative or complementary strategy to supplementation or fortification, which often face difficulties reaching rural children. An intervention strategy such as agronomic biofortification with zinc may need to be combined with other interventions to realize improvements in child nutritional status. Intervention studies are needed to determine whether there are causal links between soil and human zinc status.

## Figures and Tables

**Figure 1 nutrients-11-00221-f001:**
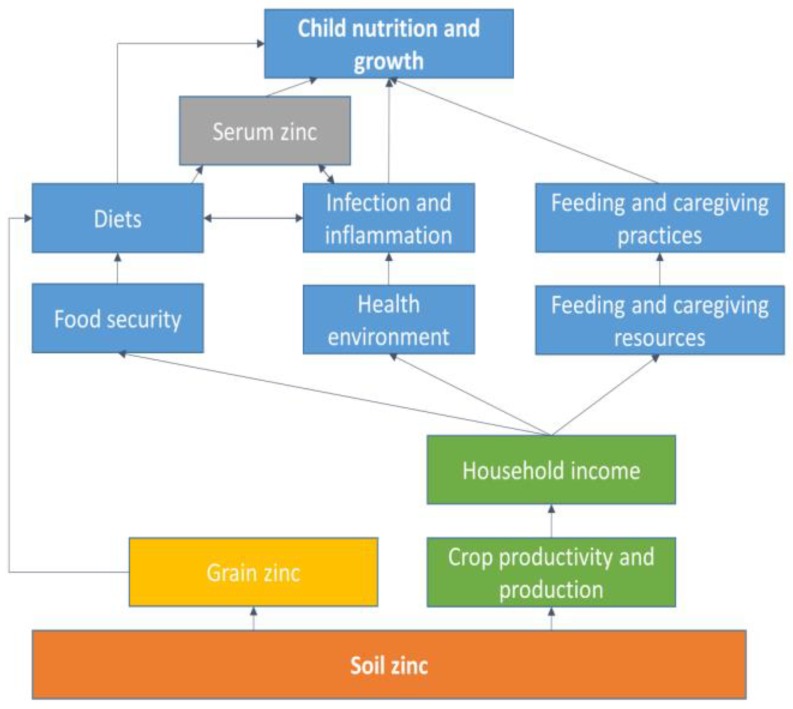
A conceptual framework describing the relationships among soil zinc, serum zinc status and linear growth of children (based on a framework to achieve optimum child nutrition and development from Black et al [26]).

**Figure 2 nutrients-11-00221-f002:**
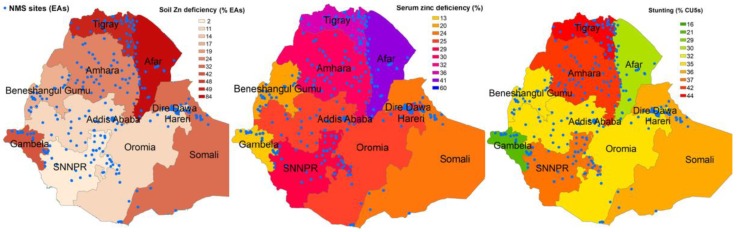
Geographical distribution of poor soil zinc, poor serum zinc status and poor linear growth of children in Ethiopia. NMS=National Micronutrient Survey; EAs = Enumeration Areas; Serum zinc deficiency (%) = Percent of under-five children who are deficient as measured by serum zinc; CU5s = children under five.

**Table 1 nutrients-11-00221-t001:** Characteristics of study participants (children under five years) from the Ethiopian National Micronutrient Survey (ENMS).

Indicators	*N*	Median (25th, 75th Percentiles) or %
Age in months	1776	36 (24, 48)
Age categories		
Age (6–11 months)	118	7%
Age (12–23 months)	288	16%
Age (24–59 months)	1370	77%
Sex (female)	1776	48%
Child had diarrhoea in preceding two weeks	1776	15%
Child received medication during the diarrheal episode	1776	5%
Child consumed meat or meat products in the last 24 h	1776	11%
Unadjusted serum zinc (μg/dL)	1171	74.1 (63.4, 87.4)
AGP (g/L)	1180	0.95 (0.75, 1.20)
CRP (mg/L)	1164	0.64 (0.25, 2.20)

**Table 2 nutrients-11-00221-t002:** Prevalence and geographical distribution of soil zinc deficiency, child serum zinc deficiency and child stunting in Ethiopia by region.

	Percentage of Households on Zinc-Deficient Soils (<1.5 mg/kg), *n* = 1298	Prevalence Serum Zinc Deficiency ^1^ (<65 μg/dL), *n* = 1171		Prevalence of Stunting (HAZ < −2.0), *n* = 1673
		Unadjusted	Adjusted ^2,3^	
**Region**				
Tigray	50	36	29	44
Afar	87	40	34	31
Amhara	25	30	28	42
Oromia	17	25	22	35
Somali	33	24	22	36
Benishangul	15	20	16	36
SNNPR	2	29	22	37
Gambella	42	13	11	21
Harari	46	32	28	29
Addis Ababa	25	60	60	16
Dire Dawa	20	29	29	32
**Age group**				
Age (6–11 months) Age (12–23 months) Age (24–59 months)	31 24 18	28 26 28	20 24 24	6 34 41
**Sex**				
Boys	20	26	23	41
Girls	20	29	25	34
**Residence**				
Urban	24	32	25	26
Rural	20	27	24	39
**National prevalence ^4^**	20	28	24	38

^1^ All subjects were non-fasting. ^2^ BRINDA internal regression correction approach, which accounts for both CRP and AGP, was applied to calculate the adjusted prevalence of zinc deficiency [32]. ^3^ Adjusted for inflammation = exp(unadjusted ln serum zinc−(regression coefficient for CRP) × (CRP− (maximum of lowest decile for CRP))−(regression coefficient for AGP) × (AGP−(maximum of lowest decile for AGP))). ^4^ Prevalence of soil zinc deficiency, serum zinc deficiency and stunting was weighted using a regional weight factor.

**Table 3 nutrients-11-00221-t003:** Spearman’s rank correlations between soil zinc, serum zinc, inflammation biomarkers child’s growth, incidence of diarrhoea in children and household food insecurity.

Indicators	Soil Zinc	Serum Zinc	AGP	CRP	HAZ	WHZ	Diarrhoea
Serum zinc (µg/dL)	0.09 **						
AGP (g/L) ^1^	0.12 **	−0.08 *					
CRP (mg/L) ^2^	0.07 *	-0.05	0.55 **				
HAZ ^3^	−0.03	0.02	0.02	−0.03			
WHZ ^4^	0.08 **	0.02	−0.01	−0.0006	−0.05 *		
Diarrhoea	0.06 **	−0.05	0.08**	−0.001	−0.01	−0.07 **	
FIES ^5^	−0.03	0.0001	0.04	−0.02	−0.02	−0.07 *	0.09 **

**p* < 0.05; ***p* < 0.01. ^1^ AGP = α-1-glycoprotein protein concentration. ^2^ CRP = C-reactive protein concentration. ^3^ HAZ = Height-for-age Z-score. ^4^ WHZ= Weight-for-height-Z-score. ^5^ FIES = The Food Insecurity Experience Scale.

**Table 4 nutrients-11-00221-t004:** Multi-level mixed linear regression model predicting serum zinc (*n* = 1171).

Fixed Effects	Estimate ^1^	SE	*p*
Soil zinc (mg/kg)	0.9	0.4	0.020
Diarrhoea in past two weeks	−1.9	1.8	0.284
CRP (mg/L) and AGP (g/L) (ref = normal)			
Elevated CRP only (mg /L)	−4.4	6.5	0.503
Elevated AGP only (g/L)	−4.1	1.4	0.003
Elevated AGP (g/L) and CRP (mg/L)	−6.8	2.0	0.0008
Age in months (Ref = 6–11 months)			
Age category 2 (12–23 months)	−1.5	3.5	0.655
Age category 3 (24–59 months)	−1.2	3.1	0.702
Sex of child (female)	−1.9	1.2	0.127
Wealth status (Ref = wealthier)			
Wealth status (poorer)	1.8	1.8	0.301
Wealth status (medium)	2.6	1.6	0.111
Time since most recent meal (hr)	−6.3	3.3	0.057
FIES ^2^	−0.1	0.2	0.693
Consumption of meat or meat products in the last 24 h	−3.0	2.0	0.136
**Random effects**			
Intercept (cluster)	29.7	11.5	0.0049

^1^ Models adjusted for regions as fixed effects. The restricted maximum likelihood (REML) method was used to estimate the parameters. ^2^ FIES = The Food Insecurity Experience Scale.

**Table 5 nutrients-11-00221-t005:** Multi-level mixed linear regression models predicting height-for-age Z-score ^1^ (*n* = 1673).

	Model 1 ^2^			Model 2 ^2^		
Fixed Effects	Estimate	SE	*p*	Estimate	SE	*p*
Soil zinc (mg/kg)	0.02	0.03	0.6035	0.02	0.03	0.522
Serum zinc (μg/dL)				−0.002	0.003	0.506
Diarrhoea in past two weeks	−0.30	0.16	0.0654	−0.3	0.2	0.091
CRP (mg/L) and AGP (g/L) (Ref = normal)						
Elevated CRP only (mg/L)	0.01	0.57	0.9842	−0.11	0.60	0.848
Elevated AGP only (g/L)	−0.09	0.13	0.4751	−0.04	0.13	0.753
Elevated CRP (mg/L) and AGP (g/L)	−0.10	0.18	0.5839	−0.05	0.19	0.781
Age in months (Ref = 6–11 months)						
Age category 2 (12–23 months)	−0.40	0.3	0.199	−38	0.32	0.235
Age category 3 (24–59 months)	−1.1	0.3	<0.0001	−1.1	0.28	<0.0001
Sex (female)	0.2	0.1	0.054	0.2	0.12	0.036
Wealth status (Ref = wealthier)						
Wealth status (poorer)	−0.2	0.2	0.214	−0.13	0.16	0.435
Wealth status (medium)	−0.2	0.1	0.269	−0.11	0.15	0.451
FIES ^3^	−0.02	0.02	0.286	−0.03	0.02	0.175
**Random effects**						
Intercept(cluster)	0.2	0.1	0.007	0.2	0.08	0.009

^1^ Models adjusted for regions as fixed effects. The restricted maximum likelihood (REML) method was used to estimate the parameters. ^2^ Model 1 adjusted for soil zinc, model 2 adjusted for soil zinc and serum zinc. ^3^ FIES = The Food Insecurity Experience Scale.

**Table 6 nutrients-11-00221-t006:** Multi-level mixed linear regression models predicting weight-for-height-Z-score^1^ (*n* = 1673).

	Model 1 ^2^			Model 2 ^2^		
Fixed Effects	Estimate	SE	*p*	Estimate	SE	*p*
Soil zinc (mg/kg)	0.05	0.02	0.026	0.05	0.023	0.045
Serum zinc (μg/dL)				0.002	0.002	0.488
Diarrhoea in past two weeks	−0.16	0.12	0.162	−0.184	0.118	0.121
CRP (mg/L) and AGP (g/L) (Ref = normal)						
Elevated CRP only (mg/L)	−0.14	0.40	0.737	−0.104	0.426	0.808
Elevated AGP only (g/L)	0.04	0.09	0.694	0.037	0.093	0.687
Elevated CRP (mg/L) and AGP (g/L)	−0.01	0.13	0.916	−0.0002	0.134	0.999
Age in months (Ref = 6–11 months)						
Age category 2(12–23 months)	−0.04	0.23	0.846	−0.020	0.228	0.931
Age category 3 (24–59 months)	0.07	0.20	0.734	0.044	0.201	0.827
Sex (female)	−0.03	0.08	0.717	−0.031	0.082	0.704
Wealth status (Ref = wealthier)						
Wealth status (poorer)	−0.40	0.11	0.001	−0.434	0.117	0.0002
Wealth status (medium)	−0.09	0.10	0.366	−0.141	0.107	0.187
FIES ^3^	0.01	0.02	0.703	0.008	0.015	0.604
**Random effects**						
Intercept(cluster)	0.12	0.05	0.005	0.124	0.048	0.005

^1^ Models adjusted for regions as fixed effects. The restricted maximum likelihood (REML) method was used to estimate the parameters. ^2^ Model 1 adjusted for soil zinc, model 2 adjusted for soil zinc and serum zinc. ^3^ FIES = The Food Insecurity Experience Scale.

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
