# Peer review of "Soil Zinc Is Associated with Serum Zinc But Not with Linear Growth of Children in Ethiopia"

_nutrients, 2019, doi:10.3390/nu11020221_

Round 1
Reviewer 1 Report
In their manuscript „Soil Zinc is Associated with Serum Zinc but not with Linear Growth of Children in Ethiopia” (nutrients-406693) the authors present an interesting association study between soil zinc, serum zinc, weight to height ratio, and linear growth in preschool children in Ethiopia. The paper is well written and follows the interesting hypothesis, summarized in figure 1, that soil zinc affects factors serum zinc and thereby possibly events such as linear growth of children. However, I have some severe concerns regarding important aspects of the study, as outlined below:
Line 114: As mentioned by the authors, serum zinc is only recommended for population level studies and only used for individuals based on a lack of a validated more suitable parameter. Therefore I can see a rationale for the more generalized type of analysis performed by the authors in the first part of their study. However, it is used as an individual marker for subsequent association studies. This needs to be critically discussed, at least.
Moreover, the number of 1776 children mentioned in the abstract is misleading, as only 1171 blood samples were available. This indicates a higher statistical power than the study actually has. Also, I am unhappy with a dataset for which a parameter of utmost importance (serum zinc) is only available for two thirds of the study group (even if an intrapolating approach is chosen).
The number of children included in the later tables (Section 3.3) needs to be specified.
Children were included from a relatively early age (6 months). As the zinc content of human milk is remarkably stable (even moderate zinc deficiency has hardly any influence) I would recommend investigating only children after a reasonable period post weaning, allowing manifestation of stunting after the actual onset of zinc deficiency. Age category 2 is certainly not suitable for this type of analysis and Age category 3 is at least questionable.
Low dietary intake may not be automatically associated with stunting as this effect results from considerable zinc deficiency. Hence, I would not expect to see a general correlation between serum zinc and height, but selective stunting in a more severely zinc deficient subgroup of the older children (above of age groups 2 and 3). I am not sure if the analytical approach is optimally suited for detecting this.
For rural communities, a correlation between soil zinc and consumed staple crops seems can be assumed, but how many children from the ENMS were actually from larger settlements (even urban areas such as Addis Ababa)? In these cases I would expect that food will have been transported from outside the AfSIS grid cells. Therefore, a comparison as in the present study would only be valid for rural communities with little trade of foodstuffs.
Figure 2 cannot be correct: The middle panel shows regions color coded according to average serum zinc in ug/dL. These values start at 13 ug/dL (which is probably lethal) and are highest at 60 g/dL, which is still below the cutoff for zinc deficiency. According to the latter I would expect far more than 50% zinc deficiency in all regions outside of Afar. Strikingly, Gambella had the lowest rate of zinc deficiency (Tab 2) despite extremely low (unphysiological) serum zinc levels. Please check if this is not actually indicating the fraction of zinc deficient individuals.
In addition to total zinc, bioavailability is a major issue. As this is mainly associated with phytate, it should be considered if there are regional differences in phytate consumption/food choices. Soil zinc is only one relevant factor, but may not even have the biggest influence.
Minor points
Line 102: “Sample weights were calculated from the 2007 census” vs. line 144 “Anthropometrics (i.e., weight and height or recumbent length) were collected…” seems to be contradictory.
Lines 197, 202, 205, 217 and 220 contain the warning: “Error! Not a valid bookmark self-reference”
Line 276: I do not agree with the statement “We found that the risk of zinc deficiency resulting from low serum zinc concentrations ….”. It indicates the zinc deficiency is a result (not indicated by) low serum zinc.
Reviewer 2 Report
The authors of the manuscript have found that the risk of zinc deficiency defined by low serum zinc concentrations was high among Ethiopian preschool children, and soil zinc was positively associated with children’s serum zinc after controlling for other factors. My specific comments are as below:
Page 3 line 114, I suggest the author remove “population-level” in the sentence, since plasma zinc is also widely used to estimate individual level zinc status in clinics. I don’t have access to reference 30 which was cited here, but since the authors used plasma zinc to assess each child zinc status, and zinc deficiency was defined as<65ug/dL here and many other publications, the author should not emphasize “population-level” here even it is what was concluded in the citation.
Page 5 line 196 and line 201 are duplicated broken sentences: The prevalence of serum zinc deficiency was comparable across (20–25%) (Error! Not a valid bookmark self-reference.), and line 205, line 217 and line 220?
The author stated they treated “cluster” as random variable and “region” as fix variable in the model. Could the authors please clarify the cluster and region variables? E.g. How many regions in total? And what the relationship between region and cluster.
For table 5 and table 6, the difference in covariates adjusted between model 1 and model 2 should be added in the footnote.
Round 2
Reviewer 1 Report
In the revised version of their manuscript „Soil Zinc is Associated with Serum Zinc but not with Linear Growth of Children in Ethiopia” (nutrients-406693) the authors have addressed all points raised in my previous review. It is unfortunate that the issue of bioavailability could not be addressed in more detail, which significantly weakens the study.